# Novel Molecular Targets for Immune Surveillance of Hepatocellular Carcinoma

**DOI:** 10.3390/cancers15143629

**Published:** 2023-07-15

**Authors:** Pietro Guerra, Andrea Martini, Patrizia Pontisso, Paolo Angeli

**Affiliations:** Unit of Internal Medicine and Hepatology (UIMH), Department of Medicine (DIMED), University of Padova, 35122 Padova, Italy; pietro.guerra@studenti.unipd.it (P.G.); patrizia@unipd.it (P.P.); pangeli@unipd.it (P.A.)

**Keywords:** hepatocellular carcinoma, hypoxia, immunotherapy, SerpinB3

## Abstract

**Simple Summary:**

Hepatocellular Carcinoma (HCC) is a common and aggressive cancer with limited treatment options. Approximately half of all patients with HCC receive systemic therapy during their disease course, particularly in the advanced stages of the disease. Immunotherapy has been paradigm-shifting for the treatment of human cancers, with strong and durable antitumor activity in a subset of patients across a variety of malignancies including HCC. However, identifying new targets is crucial to improve the effectiveness of immunotherapy. This review discusses novel molecules, cell types, and mechanisms implicated in the immunosurveillance of HCC.

**Abstract:**

Hepatocellular carcinoma (HCC) is a common and aggressive cancer with a high mortality rate. The incidence of HCC is increasing worldwide, and the lack of effective screening programs often results in delayed diagnosis, making it a challenging disease to manage. Immunotherapy has emerged as a promising treatment option for different kinds of cancers, with the potential to stimulate the immune system to target cancer cells. However, the current immunotherapeutic approaches for HCC have shown limited efficacy. Since HCC arises within a complex tumour microenvironment (TME) characterized by the presence of various immune and stromal cell types, the understanding of this interaction is crucial for the identification of effective therapy. In this review, we highlight recent advances in our understanding of the TME of HCC and the immune cells involved in anti-tumour responses, including the identification of new possible targets for immunotherapy. We illustrate a possible classification of HCC based on the tumour immune infiltration and give evidence about the role of SerpinB3, a serine protease inhibitor involved in the regulation of the immune response in different cancers.

## 1. Introduction

Hepatocellular carcinoma (HCC) is the most common primary liver cancer, followed by Cholangiocarcinoma (CCA). It is the fourth cause of cancer death worldwide and there has been an increase of 75% in its incidence in the last years [1,2]. The overall 5-year survival is 18%, making it the second most lethal tumour after pancreatic cancer [3].

HCC arises in 90% of the cases in the setting of chronic liver disease (CLD), and cirrhosis of any aetiology is its strongest risk factor [4]. There is evidence that suggests that it can also develop during the progression of Non-Alcoholic Fatty Liver Disease (NAFLD) even before the stage of cirrhosis [5,6,7].

Curative treatments, such as surgical resection or radiofrequency ablation, can be applied in the early stages [8]. Surveillance programs rely on the dosage of serum alpha-fetoprotein and the execution of abdominal ultrasound to detect early-stage tumours [9,10,11]. Nevertheless, it has been estimated that the diagnosis is incidental in 50% of cases globally [4], precluding most of the patients to undergo curative treatment. This is why there is a huge need for novel biomarkers or strategies to improve surveillance.

According to the Barcelona Clinic Liver Cancer (BCLC), the treatment algorithm suggested by the European Association for the Study of the Liver (EASL), indicates that tumours in the advanced stage must be treated with systemic therapy [12]. Accordingly, other Guidelines indicated systemic therapy as the first choice for treatment of advanced stages [13,14,15].

Since some years ago, the standard of systemic treatment was based on tyrosine kinase inhibitors (Sorafenib, Lenvatinib, Cabozantinib or Regorafenib). In addition, new drugs and natural compounds have been proposed, such as sanguinarine, targeting HIF-1α and TGF-β signalling pathways, which leads to subsequent EMT and cell migration inhibition [16]. Several studies have revealed indeed that sanguinarine impedes tumour metastasis, however, its low chemical stability and poor oral bioavailability remain key issues that are currently approached using novel release methods and developing alternative analogues [17].

Over the past decade, immuno-oncology has taken the lead in the treatment of malignancies including liver cancer and consequently, several new drugs have been approved, including anti-VEGF and Immune Checkpoint Inhibitors (ICI) [18].

Despite the initial success of the new therapies, evidence suggests that there are different problems to overcome: the response rate of ICI monotherapy is 20%, while combination therapy reaches approximately 30–36% [19,20,21]. One of the mechanisms of resistance to ICI therapy can be the low expression or absence of immune checkpoints [22]. In other kinds of solid tumours, including breast, lung and oesophageal cancer, screening for the presence of PD-L1 is an effective way to predict the response to the therapy [23,24,25]. Unfortunately, due to the complexity of HCC pathogenesis and microenvironment, this approach did not achieve the same results for hepatocellular carcinoma.

Still, clinical trials, which have demonstrated the efficacy of immunotherapeutic agents, have found interesting differences and provided possible hypotheses to overcome this problem.

In the IMbrave150 phase III clinical trial Atezolizumab, an anti-programmed death ligand 1 (PD-L1) monoclonal antibody, in association with Bevacizumab, an anti-vascular endothelial growth factor (VEGF) monoclonal antibody, has been shown superior to Sorafenib [26]. More recently, according to the results of the HIMALAYA trial, the association of Tremelimumab, an anti-cytotoxic T lymphocyte-associated antigen 4 (CTLA-4) monoclonal antibody, and Durvalumab (anti-PD-L1 monoclonal antibody) have been added to the possible first-line strategy of systemic therapy [27].

Subgroup analysis of clinical trials has uncovered an important difference in the response to ICI therapy depending on the aetiology of CLD and HCC [28]. In the IMbrave150, patients with non-viral aetiology did not show benefits of Atezolizumab and Bevacizumab over Sorafenib treatment (HR for death, 1.05; 95% CI, 0.68–1.63) [29]. On the other hand, in the HIMALAYA trial, the superiority of Tremelimumab and Durvalumab was confirmed for HBV-infected and non-viral patients, but not for HCV-infected patients (HR, 1.06; 95% CI, 0.76–1.49) [30].

Moreover, in the KEYNOTE-240 trial, comparing Pembrolizumab (anti-PD-L1 monoclonal antibody,) vs. placebo in patients pre-treated with Sorafenib, the overall survival (OS) was significantly higher in patients with HBV infection than in non-viral or HCV-related CLD [31]. However, in the CheckMate 450 trial, comparing Nivolumab, an anti-programmed death 1 (PD1) monoclonal antibody to Sorafenib as first-line treatment, HBV and HCV-infected patients showed a better response than non-viral patients, although the difference was not statistically significant [32]. Finally, the LEAP-002 trial, comparing Pembrolizumab vs. Lenvatinib did not show any difference according to aetiology [33].

These conflicting results should be taken with caution since they originate from post hoc analysis and the number of patients included was quite small. However, it is important to consider that when analysing the response to therapy, the aetiology of CLD and HCC is an important factor.

On this line, several hypotheses have been provided. For example, in preclinical models of Non-Alcoholic Steatohepatitis (NASH) related HCC an accumulation of exhausted CD8+PD-1+ cells has been reported [34]. Although treatment with anti-PD-1 led to an increase in CD8+ infiltration, tumour regression was not achieved. This finding is likely since NASH-related CLD causes aberrant T-cell activation and impaired immune surveillance.

A second proposed mechanism of resistance to immunotherapy is the presence of strong inhibition of the immune response by the Tumour Microenvironment (TME) [21].

These pieces of evidence taken together indicate that a thorough study of the immune system in HCC is required to better understand HCC development and the possible response to therapeutic strategies based on Immune Checkpoint Inhibitors. In this review, we will analyse novel molecules, cell types and mechanisms implicated in the immunosurveillance of HCC.

## 2. The Immuno System in HCC

### 2.1. Antigen Recognition

During carcinogenesis, tumour cells can be detected by the immune system for the presence of different antigens. Two different kinds of tumour-associated antigens (TAA) are recognized: non-mutated self-antigens, which are Major Histocompatibility Complex I (MHC-I) restricted antigens, and Major Histocompatibility Complex II (MHC-II) restricted antigens, which arise from non-synonymous somatic mutations [35].

Dysregulation in the antigen-presenting process is one of the possible mechanisms of resistance to Immunotherapy. It has been demonstrated that high MHC-II levels in tumours correlate with better response to ICI [36]. Moreover, mutations of beta2-microglobulin lead to reduced MHC-I expression and impaired antigen presentation, affecting antitumour response [37].

In HCC, different non-mutated self-antigens have been identified. CD8+ T cells directed against alpha-fetoprotein (AFP), glypican 3 (GPC3), melanoma-associated antigen 1 (MAGEA1), and New York oesophageal squamous cell carcinoma 1 (NY-ESO1) were found in the serum of patients with HCC [38].

Tumour-specific antigens (TSA) can be useful in different cancer-directed therapies. Given the fact that they are expressed only in tumoral cells, they can be used to target therapies selectively against cancer cells. In particular, GPC3 and AFP have been studied to develop vaccines to bust immunity against cancer cells. Several trials are ongoing either alone or in combination with other immunotherapies [39,40]. Vaccines can also be built using dendritic cells (DCs), specialized antigen-presenting cells (APCs) that can activate T and Natural Killer (NK) cells and produce a cytotoxic T-lymphocyte (CTL) response.

On the other hand, during hepatocarcinogenesis, different genomic mutations can occur, creating cancer neoantigens. Most of the new peptides arising from non-synonymous somatic mutations are rarely shared by other tumours [41].

For different kinds of tumours, the sum of these mutations (Tumour Mutational Burden–TMB) is frequently used as a surrogate of the number of neoantigens and related to the probability of the immune system identifying and eliminating cancer cells, and thus, to the probability of response to immunotherapy [42]. However, in HCC genomic analysis, a low prevalence of high TMB was identified [43].

### 2.2. Tumour Microenvironment

It is well known that the liver is an immunotolerant organ. Through the portal vein system, the liver receives blood drainage by the intestine, thus it is constantly exposed to antigens and toxins derived from gut microbiota. The equilibrium that controls the activation and suppression of the immune system must be considered for a thorough investigation of cancer development and response to ICI [44].

Of note, multiple studies on the Tumour Microenvironment (TME) of several cancers, including HCC, have given contradictory results and this could be ascribed to the different roles that every actor can have in this dialogue, depending on its different state of activation [42].

The immune response directed against cancer is predominantly cytotoxic and it is carried out by CD8+ T lymphocytes and NK cells. It has been proven that a high concentration of CD8+ or NK in the TME is associated with longer survival and better response to therapy [18].

Different mechanisms are involved and stimulated by tumour cells to inhibit CTL response and immunotherapy aims to counteract this inhibition and boost the immune response.

Hereafter we will summarize the various players of the tumour microenvironment and the different strategies studied to boost the antitumoral activity. (Table 1).

#### 2.2.1. Adoptive Cell Therapy

Due to the presence of tumour-associated antigens, effector cells such as T cells or NK cells can be engineered to increase their cytotoxic ability. The two major fields of research are the Chimeric Antigen Receptor (CAR) T cells and T Cell Receptor (TCR) Engineered T cells. 

The first one consists of T cells expressing a receptor composed of an extracellular domain containing a single-chain fragment of a monoclonal antibody specific for TAA, a transmembrane domain and a signalling domain. Recognition of antigens is not MHC-restricted, thus this technology can be applied in different situations [45].

Preclinical models have demonstrated the efficacy of CAR T cells directed against GPC3. It was demonstrated that anti-GPC3-CAR T cells effectively eliminate tumours in patient-derived xenograft murine models of HCC [53]. Recently, it has been found that Human-VH-CAR T cells targeting GPC3 lead to tumour eradication in in vitro and in vivo models [54]. Moreover, the efficacy of combining anti-GPC3-CAR T cells and Sorafenib has been proven in vivo in both immunocompetent and immunodeficient mice [55].

Based on these experimental results, different clinical trials are ongoing aiming to demonstrate the safety and efficacy of this novel therapeutic approach (NCT03130712, NCT02715362, NCT03084380). In 2020, a phase I trial (NCT02395250, NCT03146234) studied the safety of anti-GPC3-CAR T cells in 13 patients in the advanced stage of HCC. None of the patients experienced grade 3 or 4 neurotoxicity, however, 9 out of 13 patients developed cytokine release syndrome (CRS), a reaction that remains a major problem of CAR-T therapy, since it can lead to life-threatening conditions [56].

T cell receptor transgenic T cells consist of engineered T cells with high-affinity TCR specific for tumoral antigens and, at variance from CAR T, these cells bear an antigen recognition capacity limited to MHC [41,45]. In 2015, genetically modified T cells with a TCR specific for HBsAg (HBV-TCR-T-cells) were generated and injected in an HBV-positive patient, who underwent liver transplantation for HCC and developed metastasis after the transplant. HBV-TCR-T-cells were able to selectively recognize metastatic cells without affecting the HBV-negative transplanted liver [57]. Since the major problem of using this therapeutic strategy in transplanted patients is the administration of immune suppressive drugs, some years later the same group managed to engineer HBV-TCR-T-cells conferring resistance to Tacrolimus and Mycophenolate [58]. Moreover, HBV-TCR-T-cell immunotherapy has been proven safe in a dose escalation, phase I clinical trial (NCT03899415) [59].

Similar results were obtained using engineered T cells, directed against HCV antigens where recognition of HCC cells expressing HCV NS3 protein was achieved in vitro and the regression of the tumour mass of an HCV-positive HCC was reported in a murine model [60].

T Cell Receptor engineering has been employed not only against viral antigens but also against tumoral proteins, such as AFP. Preclinical studies have indeed confirmed the efficacy of AFP-specific transgenic T cells against liver cancer cells and proven their efficacy in killing HCC cells in vitro and in vivo [61,62].

Different clinical trials are ongoing to assess the safety of TCR-engineered T cells (NCT02869217, NCT03132792, NCT03159585, NCT02686372, NCT01967823, NCT05195294, NCT03638206, NCT04745403, NCT05339321, NCT02719782, NCT04368182) and the results will be available soon.

#### 2.2.2. Lymphoid Inhibiting Cells

##### T Regulatory Lymphocytes

T regulatory (Treg) cells are T lymphocytes (CD4+ CD25+ FoxP3+) that have been proven to inhibit the cytotoxic response and their presence is associated with poor prognosis [63].

Their differentiation is stimulated by Transforming Growth Factor (TGF)-beta and Interleukin (IL)-10 [64] produced by tumour-associated macrophages and other types of cells of the tumour microenvironment. They down-regulate the expression of CD 80 and CD 86 costimulatory molecules in APCs and decrease the CTL response producing TGF-beta and IL-10 [43].

Hypoxia is a widely studied condition in tumours, especially in HCC, often arising in livers with cirrhosis, where sinusoid capillarization determines impaired oxygen diffusion [65]. In this condition, which has been associated with a worse prognosis, immunosuppressive TME features have been described [66,67]. In particular, it has been demonstrated that hypoxia can stimulate the recruitment of Treg cells, supported by the activation of the Hypoxia-Inducible Factor (HIF) which leads to the production of chemokine (C-C-motif) ligand 28 (CCL28), engaging Treg cells [68].

Recently, a hypoxia-driven interaction between Tregs and dendritic cells has been described. An increased concentration of Treg and CD8+ exhausted T cells were indeed found in hypoxic regions of HCC, associated with an increased amount of M2 macrophages and a DCs subset with low Human Leukocyte Antigen (HLA)-DR expression, consistent with a reduction of antigen-presenting function. This particular DCs phenotype seems to be associated with Tregs, suggesting that hypoxic conditions can result in activation of Tregs, exhaustion of CD8+ cells and reduction of DCs antigen-presenting functions [69].

#### 2.2.3. Myeloid Inhibiting Cells

##### Tumour-Associated Macrophages (TAMs)

TAMs are an important component of the TME, exerting a role in the inflammatory response in HCC [70]. They can be classified into two different categories: M1 and M2. 

M1-TAMs activate the inflammatory response producing Tumour Necrosis Factor (TNF)-alpha, chemokines and interleukins, they can also release Reactive Oxygen Species (ROS) and activate T Helper (TH) 1 lymphocytes by presenting different antigens through MCH-II [47].

On the other hand, M2-TAMs are associated with anti-inflammatory activity. Polarization to M2-type happens after secretion of TGF-beta, IL-4, IL-10 and IL-13 by TH2 and tumour cells. They can promote tumour growth, metastatic spread and immune escape and are associated with increased tumour nodules and vascular invasion [71].

Due to their fundamental role in HCC development and immune-escaping, different strategies to inhibit M2-TAMs have been studied. Cheng et al. [47] have recently reviewed the different approaches that have been developed to block the interaction between TAMs and HCC cells and these strategies have been divided into three different categories: (a) cutting off the source and eliminating the production of M2 TAMs, (b) remodelling M2 TAMs to M1 TAMs and (c) blocking communication between M2 TAMs and liver cancer cells. 

The immune impairment determined by hypoxia is exerted also through its effect on TAMs function. It has been reported that HIF-1alpha, which is stimulated by hypoxic conditions, up-regulates macrophage expression of Triggering Receptor Expressed on Myeloid cells-1 (TREM-1). TREM-1-positive TAMs accumulation results in CD8+ inhibition and enhanced chemotaxis of Tregs throughout secretion of Chemokine (c-c motif) Ligand 20 (CCL20). This occurrence leads to resistance to anti-PDL-1 therapy, supported by the fact that the blockage of TREM-1-positive TAMs leads to a restoration of anti-PDL-1 efficacy [72].

A recent study has identified another interesting effect of hypoxia on macrophages. Using spatial transcriptomics and single-cell data, the authors have identified a subpopulation of macrophages SPP1+, which, when stimulated by hypoxic conditions and HIFs pathways activation, can form a tumour immune barrier (TIB) that is associated with ICI resistance. Targeting this TIB promotes CD8+ T cell infiltration into the tumour [73]. 

##### Myeloid-Derived Suppressor Cells (MDSCs)

MDCSs are immunosuppressing immature cells with APC functions [74]. They have been associated with a reduction in T cell proliferation, an enhancement of Tregs function and the production of VEGF [18]. They are associated with poor prognosis when accumulated in cancer tissues, and their depletion in peripheral blood is associated with a restoration of cytotoxic function [75].

Recently, this cell phenotype has been studied for targeted therapy, and phosphodiesterase 5 (PDE5) inhibition has been proven to reduce MDSCs function in different kinds of cancer as multiple myeloma, head and neck cancer, colon cancer and lymphocyte leukaemia [48]. In line with these results, a decrease of antioxidant enzymes was demonstrated in a murine model of HCC induced by alcohol and aflatoxin administration and these enzymes were restored after treatment with Sildenafil or Tadalafil (two PDE5 inhibitors) [76]. One year later the same group reported that HCC treatment with a PDE5 inhibitor leads to transaminases and cholestasis enzymes restoration in serum, associated with normalization of mRNA expression of cancer-associated transcription factors such as c-Myc or Akt, of TGF-beta and angiogenesis markers such as HIF-1alpha and VEGF [77].

At the same time, another team [78] found out that Tadalafil is effective in suppressing tumour progression and enhancing immune response by implementing the ratio of CD8+ activated cells in a murine model of plasmid-induced HCC. This PDE5 inhibitor was administrated in combination with JQ-1, a bromodomain and extra terminal (BET) proteins inhibitor which has been proven effective in several tumours but not HCC.

##### Dendritic Cells

DCs are APCs that physiologically activate the T-guided immune response through the MHC-II antigen presentation. They can differentiate in a regulatory subtype (CD14+CTLA4+) [79] able to inhibit the CTL through the production of IL-10 and indoleamine-2,3-dioxygenase [80]. 

Given their ability to stimulate the CTL response, DCs are currently studied for vaccine therapy using specific tumour antigens such as AFP, GPC-3 and MAGEA-1, or total lysate-pulsed. In 2005, a clinical trial was carried out in 31 patients with advanced HCC, who were treated with DCs pulsed with autologous tumour lysate and no significant adverse events were registered using this approach [81].

Some years later, a phase II clinical trial was conducted in a group of 35 patients with advanced HCC who were treated with DC pulsed with autologous tumour lysate. No toxicity or evidence of autoimmunity was observed. Evidence of one partial radiological response was reported, while 6 additional patients presented stable radiological disease, 4 of which were associated with a reduction of AFP serum levels [82].

In 2012 another DCs-based vaccine was developed by pulsing DCs with a cocktail of TAA, namely α-fetoprotein, glypican-3 and MAGEA-1. A phase I/II clinical trial enrolling 5 patients demonstrated a good tolerance in all patients, however, clinical response was observed only in one patient [83].

In the same year, another autologous pulsed DCs vaccine was developed. The study was conducted on thirty patients with advanced HCC, divided into two groups, one receiving an autologous pulsed DC vaccine, and the other the best supportive care. An improvement in survival between the two groups was observed, with a median overall survival of 7 months (mean 9.8 ± 7.8 months) in the treated group, compared to 4 months in the control group (mean 5.2 ± 2.6 months) [84].

In 2015, a DC vaccine using autologous tumour stem cells (TC) was developed. Eight HBV-positive patients with HCC stage of BCLC A, with a liver tumour greater than 5 cm, or of BCLC B were enrolled. These patients underwent surgical resection after six weeks of leukapheresis to obtain peripheral DCs and one-week later Trans Arterial Chemoembolization (TACE) was performed, followed by three injections of the DC-TC vaccine. This approach was safe, without evidence of exacerbation of HBV hepatitis [85].

At the same time, a phase I/IIa trial of adjuvant therapy with a DCs vaccine was carried out. Twelve patients, who underwent successful curative treatment for HCC, were injected with DCs vaccines pulsed with AFP, GPC3 and MAGEA1. None of the patients experienced adverse reactions of grade greater than 2 and the median time to progression in the treated group was three times greater than that observed in the control group (36.6 vs. 11.8 months) [86].

In addition, in 2017 DCs vaccine therapy was combined with radiotherapy in a phase I clinical trial in patients with advanced HCC that developed more frequently a grade II bone marrow suppression, compared to controls [87].

Another important mechanism that needs attention when considering the role of DCs in HCC is their relation with beta-catenin activation, found in a patient with resistance to anti-PD-1 and PD-L1 treatment [88]. Recently, the mechanism that drives the immune impairment in beta-catenin overactivation has been uncovered [89], consisting of an impairment in the recruitment of CD103+ DCs, which are associated with effective anti-tumour response [90]. Moreover, the lack of CD103+ DCs is also associated with a lack of tumour-specific CD8+ T cells, consistent with the hypothesis of an immune-exclusion mechanism. The key player of this effect seems to be chemokine (C-C-motif) ligand 5 (CCL5) since it was found down-regulated in the beta-catenin mutant HCC, while its up-regulation led to restoration in DC infiltration and activation.

Of note, it has been recently found that in 70–85% of liver tumours carrying the CTNNB1 mutation, which results in a gain-of-function of beta-catenin, there is an important reduction in the immune infiltrate, probably due to a lack in the CCL5-guided chemotaxis. In the remaining 15–30% of the cases, CCL5 was overexpressed leading to an accumulation of CD8+ T cells infiltration, likely as a consequence of the activation of other APC-related genes [91].

#### 2.2.4. Non-Parenchymal Hepatic Cells

The immunotolerant characteristics of the liver are mainly elicited in homeostasis by stromal cells such as Kupffer cells (KC), hepatic stellate cells (HSCs) and liver sinusoidal endothelial cells (LSECs) [92].

Kupffer cells are liver-resident macrophages, with functions similar to those of monocyte-derived macrophages, but derived from erythromyeloid progenitor cells from the yolk sack [93].

Liver sinusoidal endothelial cells are APCs able to activate CTL and recruitment of lymphocytes [94].

In physiological homeostasis conditions, KCs and LSECs stimulate the Tregs response via the secretion of anti-inflammatory cytokines such as IL-10 and IL-4, however, in case of liver damage, they can activate inflammation through the secretion of IL-1β, IL-6, IL-12, IL-18, and TNF-α [44,95]. KCs turnover has been proven to be deficient in Non-Alcoholic Fatty Liver Disease (NAFLD), where instead activated macrophages were detected, leading to progression to steatohepatitis [96,97].

Some therapeutic strategies are addressed to inhibit KCs-dependent macrophage accumulation, such as the inhibition of the chemokine (C-C-motif) ligand 2 (CCL2)/chemokine (C-C-motif) receptor 2 (CCR2) axis. This therapeutic target has been studied in NAFLD progression [98,99], but also in HCC development [50]. CCL2 was found overexpressed in human HCCs and its expression was negatively correlated with prognosis. Moreover, CCL2/CCR2 inhibition determined a reduction of tumour growth, metastatic spread and post-surgical recurrence in preclinical models [50].

Recently, another important mechanism that implicates KCs in CLD progression and HCC development has been uncovered. KCs were proven to have a pivotal role in platelet recruitment during different NASH stages, and platelet accumulation was associated with the worsening of NAFLD and HCC development [100]. Accordingly, a recent survey of patients with chronic viral hepatitis revealed an association between the use of low-dose aspirin and a lower risk of HCC development and liver-related mortality, without a higher risk of bleeding [51].

Hepatic stellate cells are the most important cells in generating liver fibrosis in response to external damage such as viral infection, NASH or alcohol [101]. HCC arises in livers with cirrhosis in the majority of cases, but the relation between HSC fibrogenesis and HCC is still poorly understood. 

Hepatic stellate cells can exert different functions in HCC development, depending on the disease stage. Cytokine- and growth-factor-expressing HSCs (cyHSCs), upregulated in the initial stages of fibrosis and cirrhosis, are enriched in hepatocyte growth factors and can protect against tumour development. On the other hand, myofibroblastic HSCs (myHSCs), proliferate in advanced stages of liver disease and these cells synthesize Collagen I, whose deposition leads to liver stiffness and tumour-promoting activation of the TAZ pathway [102].

## 3. Immune Classification

Given the important role of the immune system in HCC development and response to therapy, the following classification of HCC, based on tumour immune characteristics has been recently proposed.

### 3.1. The Inflamed Class: Immune-Active, Immune-Exhausted and Immune-like

Sia et al. [103] reported for the first-time evidence that in a group of 228 patients, up to 25% had an increase in markers of the inflammatory response, increased cytolytic activity and enhanced PD1 and PDL1 expression. They termed this group the immune class and later defined the inflamed class. 

The inflamed class was then divided into two subclasses: the immune-active and the immune-exhausted. The first one is characterized by the presence of Interferon (INF) signature, T cells and adaptive immune response gene expression, while the second one is dominated by TGF-beta signalling and other immunosuppressive components. Interestingly, the two classes do not differ in the immune infiltration, Tertiary Lymphoid Structures (TLS) number, PD-L1 and PD-1 expression. The immune-active class demonstrated to have lower recurrence after resection, and better survival and was found as an independent prognostic factor of overall survival, suggesting that the state of activation of the immune system plays an important role in defining the TME role and tumour behaviour. 

Since the inflamed class included only 25% of the HCCs analysed in the first study, new subclasses were identified in the following study [91]. A group of tumours with characteristics similar to those of the inflamed class was identified and classified as an “immune-like class”. This subgroup, presenting clinical and pathological characteristics similar to those of the immune classes, was characterized by high INF signalling, immune infiltration, M1 macrophage differentiation and remarkable activation of Wnt-beta-catenin signalling. 

### 3.2. The Non-Inflamed Class: Immune-Intermediate and Immune-Excluded

The non-inflamed class was identified by Montironi and colleagues [91] and comprises the remaining 63% of liver tumours that do not fulfil the criteria to enter into the inflamed class. It can be divided into immune-intermediate and immune-excluded, based on the mechanism of immune evasion.

The immune-intermediate class is characterized by a decrease in immune infiltration associated with TP53 mutations and higher levels of deletions in genes related to INF signalling and antigen presentation.

The immune-excluded class is characterized by the lowest level of immune infiltration, associated with a high frequency of CNNTB1 mutation, causing an activation of the Wnt-beta-catenin pathway that can inhibit leukocyte migration, thus causing a profound suppression of the immune response, as described previously.

This classification could be clinically useful for different reasons. First of all, the inflamed class was characterized by the presence of two inflammatory signatures predicting ICI response, although additional studies are needed to prove this correlation. Secondly, the possibility of identifying each class signature using liquid biopsy might be useful in a clinical setting to avoid invasive procedures. 

## 4. Role of SerpinB3 in the Immunosurveillance of HCC

Recent studies have identified SerpinB3 as a new immunomodulator in different cancers, including HCC.

SerpinB3, previously known as Squamous Cell Carcinoma Antigen–1 (SCCA-1), is a member of the serine protease inhibitors (SERPINs), that is physiologically expressed in the basal and parabasal layers of normal squamous epithelium [104].

This serpin was found overexpressed in different tumours of epithelial origins, such as head and neck, cervix and uterus, lung [105,106] and also in primary liver tumours, including HCC [107], cholangiocarcinoma [108], and hepatoblastoma [109], where its detection was associated with poor prognosis and early recurrence after therapy. While SerpinB3 is undetectable in normal livers, its expression progressively increases during the evolution of chronic liver disease and HCC development [107,110]. Despite the protective effect of SerpinB3 in acute damage [111], its chronic expression in response to oxidative stress conditions plays a pivotal role in oncogenesis and in particular in HCC, affecting different cell activities that led to resistance to apoptotic cell death [112,113,114,115], increased cell proliferation and invasion [116].

Recently, this molecule has been also related to immune infiltration and the promotion of an immunosuppressive TME in other kinds of cancer. Of note, a study investigating its role in cervical cancer found that high expression of SerpinB3 was related to immune-exhausted TME and a higher risk of tumour progression and recurrence. Evidence suggested that this role could be mediated via STAT-dependent secretion of chemokine recruiting MDSC and M2 macrophages [117].

Hereafter we will briefly review the role of this molecule in HCC immunomodulation, as described in Figure 1.

TGF-beta, which is one of the major cytokines involved in impaired immune response, is induced by SerpinB3 and this effect requires the anti-protease activity of this serpin, since deletions in the reactive site loop of the serpin abrogate this effect [118], while a single amino acid substitution, found in the polymorphic variant SCCA-PD determines a gain of function [119].

In a murine model of lung fibrosis, it was interesting to note that mice transgenic for SerpinB3 showed more collagen deposition, but lower inflammatory infiltration than the wild-type control mice [120,121].

As reported before, beta-catenin activation is associated with a worse prognosis in HCC since it inhibits the migration and activation of DCs resulting in an impairment of the immune response. Of note, SerpinB3 and TGF-beta expression have been related to the WNT-beta-catenin pathway activation in patients with HCC and those with this particular pattern presented earlier tumour recurrence and worse prognosis [107]. The positive relation between SerpinB3 and beta-catenin, associated with more aggressive tumours, was also reported in colorectal cancer [122]. 

Moreover, a recent study found evidence that SerpinB3 can upregulate the WNT signalling by upregulating the expression of the low-density lipoprotein receptor-related protein (LPR) family. In particular LPR-1, -5 and -6 were upregulated by SerpinB3 and this resulted in an increased beta-catenin translocation into the nucleus. Interestingly, this activity was found mediated by a particular cellular membrane binding site which is shared by other SERPIN family members such as alpha1-antitrypsin and nexin-1 [123].

In line with these results, in a murine model of NASH, mice that were genetically modified and expressed SerpinB3 deleted in the reactive site loop, showed less TGF-beta expression, associated with reduced macrophage infiltrate in the liver. On the other hand, transgenic mice overexpressing SerpinB3 presented not only higher expression of TGF-beta but also a significant increase of macrophage infiltration, characterized by overexpression of TREM2 [124]. This peculiar macrophage phenotype has been associated with the severity of steatosis, inflammation, hepatocyte ballooning, and fibrosis [125]. 

The immune-suppressive effect of SerpinB3, determining an impairment of the cytotoxic immune response was also described in oesophageal cancer, where SerpinB3 was inversely correlated to markers of immune activation, namely CD80, CD86, TLR4, and CD38 within the tumour, and with peritumoral expressions of CD8+ and CD107+ cells [126].

These results could be determined by the induction of Tregs by SerpinB3, as found in a murine model of Systemic Lupus Erythematosus where the administration of the serpin determined an increase of Tregs in the spleen, promoting a more tolerogenic immune phenotype and slower disease progression [127].

Of note, TGF-beta has been proven in strict correlation with hypoxia and HIF-1alpha secretion, besides negatively affecting the immune response. This pathway can stimulate EMT in HCC cells and is related to the development of metastasis, gaining particular interest in a therapeutic approach [16,17].

### Response to Hypoxic Conditions and HIF Induction

Tumour hypoxia is found to be an independent negative prognostic factor in many solid tumours, including HCC, associated with metastatic spread and therapy escape [128,129]. It was proven to promote angiogenesis, epithelial-to-mesenchymal transition (EMT), and invasiveness in tumoral cells [130,131,132,133,134].

HIF-1alpha and -2alpha is the main intracellular effector of response to hypoxia. The first one is associated with cell proliferation, metabolic changes, angiogenesis, invasion, and metastasis [130,135,136,137,138,139]. The second one stimulates cell proliferation by promoting c-Myc activation, radio- and chemo-resistance, self-renewal capability and stem cell phenotype in non-stem cell populations [140,141,142,143,144,145].

In a recent study, it has been demonstrated that in a murine model of HCC, mice with HIF-2alpha specific deletion show a significant reduction in tumour nodules compared with the control group and a strong association between HIF-2alpha and SerpinB3 expression was observed and confirmed in human specimens of HCC [133]. These findings are in agreement with previous studies indicating that HIF-2alpha activates the transcription of SerpinB3 through the binding to its promoter [146]. Moreover, a positive correlation between HIF-2alpha and oncogenic signals such as YAP and c-Myc was observed, while these pathways were not activated in mice with HIF-2alpha deletion [133]. Accordingly, SerpinB3 was found to be able to increase c-Myc expression by inhibiting its degradation and by stimulating the Hippo pathway through an enhanced expression of YAP [147].

The interaction of SerpinB3 with the HIF proteins has been well defined in a recent study demonstrating that SerpinB3 can act as a paracrine mediator up-regulating HIF 1 and 2alpha [148]. While HIF 1-alpha is rapidly upregulated by SerpinB3 at the transcriptional level, promoting cell survival and angiogenesis, HIF 2-alpha is stabilized by SerpinB3 that prevents its proteasomal degradation, via selective NEDDylation, promoting cell proliferation and favouring HCC progression in a positive loop with SerpinB3 induction [148].

These findings indicate that SerpinB3 is profoundly involved in the pro-oncogenic effects described in hypoxic conditions and mediated by HIFs.

## 5. Conclusions

Immunotherapy is a novel and important field to be considered in the treatment of HCC. Despite different mechanisms that the tumour activates to induce immunotolerance and suppress the immune response against cancer, different therapeutic strategies have been developed, especially targeting different components of the tumoral microenvironment, to restore an efficient immune response. In this setting, targeting both the reactive site loop of SerpinB3 and/or blocking its interaction with the LRP co-receptors could become novel therapeutic strategies which can control not only tumour cell growth and spreading, but also the restoration of effective immune surveillance within the tumoral microenvironment.

## Figures and Tables

**Figure 1 cancers-15-03629-f001:**
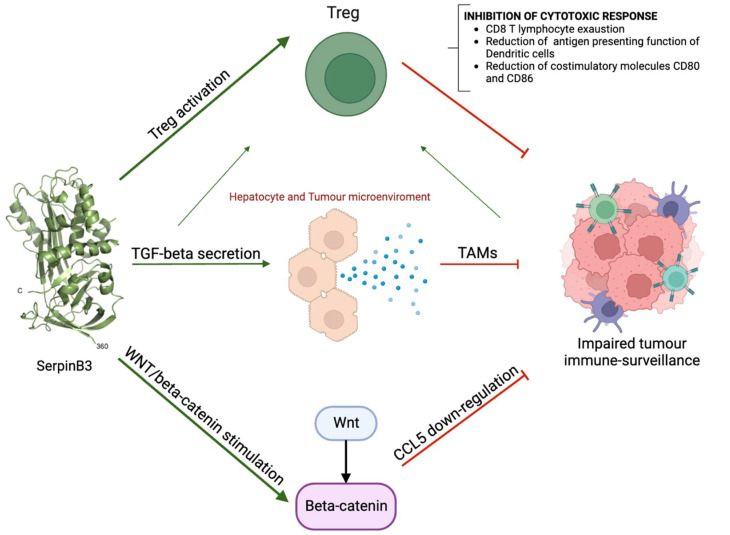
The multiple mechanisms through which SerpinB3 determines immunodepression. TAMs, Tumour Associated Macrophages.

**Table 1 cancers-15-03629-t001:** Different immunotherapy strategies for treating HCC.

Target Cell		Mechanism of Surveillance	Suggested Therapy	References
Adoptive Cell Therapy			CAR-T	[45]
			TCR Engineered T cells	[41]
Lymphoid cells	Tregs	Secretion of TGF-beta and IL-10. Inhibition of M1 activity.	Immune Checkpoint Inhibitors	[46]
Myeloid cells	TAMs	Inhibition of CD8+. Stimulation of Tregs. Production of proangiogenic and pro-proliferation cytokines	Eliminating production	[47]
			Remodelling M2 polarization	
			Blocking communication with cancer	
	MDSCs	L-Arginine and nitric oxide synthase-2 pathway-dependent inhibition of cytotoxic response	PDE5 inhibitors	[48]
	DCs	Antigen-presenting function. Boosting cytotoxic activity	Vaccine strategies	[49]
Non-parenchymal cells	KCs	Macrophages chemotaxis and polarization. Up-regulation of PD-1/PD-L1.	Blocking CCL2/CCR2 axis	[50]
			Platelets inhibition	[51]
	HSCs	Promotion of fibrosis and oncogenesis	Targeting molecular pathways of activation (e.g., TGF-beta)	[52]

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
