# Peer review of "Novel Molecular Targets for Immune Surveillance of Hepatocellular Carcinoma"

_cancers, 2023, doi:10.3390/cancers15143629_

Round 1

Reviewer 1 Report

Journal of Cancers

Review Article;

The article entitled “Novel molecular targets for immune surveillance of hepatocellular carcinoma”. The author reviewed Hepatocellular carcinoma targets. As the incidence of Hepatocellular carcinoma is increasing and lack of effective screening programs often results in delayed diagnosis, making it a challenging disease to manage. Immunotherapy has emerged as a promising treatment option for different kinds of cancers, with the potential to stimulate the immune system to target cancer cells. The current immunotherapeutic approaches for Hepatocellular carcinoma have shown limited efficacy. Since Hepatocellular carcinoma arises within a complex tumour microenvironment characterized by the presence of various immune and stromal cell types, the understanding of this interaction is crucial for the identification of effective therapy. The  illustrate of possible classification of Hepatocellular carcinoma based on the tumour immune infiltration gives evidence about the role of SerpinB3, a serine protease inhibitor involved in the regulation of the immune response in different cancers.

I carefully read the manuscript and found it is a wonderful effort by the author to review molecular targets for immune surveillance of hepatocellular carcinoma. But there is some minor revision needs and fulfill the mistake which the author could have done during writing. The article could be considered for publication in the prestigious Cancers Journal.

Comments for Authors

Ø  Write the keyword words in alphabetical order.

Ø  “Introduction Section” The author needs to explain more the introduction section and include more references.

Ø  The author needs to include natural agents targeting Hepatocellular carcinoma immune surveillance.

Ø  It will be more interesting to include a detailed table of target agents and the mechanism of Hepatocellular carcinoma immune surveillance.

Ø  It will be better to discuss about drug resistance in Hepatocellular carcinoma immune surveillance.

Ø  The manuscript needs critical revision and correct grammatical mistakes.  

Ø  The author targets SerpinB3 very often during the study. Could the author through light on another target Hepatocellular carcinoma immune surveillance?

Cite the following references;

v  DOI: 10.2174/1871520622666220831124321

v  doi.org/10.1038/s41419-019-2173-1

Author Response

We are very pleased with the comments of the reviewer and we would like to thank you for your thorough indications and suggestions.

  • Write the keyword words in alphabetical order:

The keywords have been written in alphabetical order.

  • “Introduction Section” The author needs to explain more the introduction section and include more references:

The introduction section has been expanded with more data on epidemiology, treatment and screening for HCC. We are grateful to the reviewer for his suggestion that improve the manuscript.

  • The author needs to include natural agents targeting Hepatocellular carcinoma immune surveillance:

We are deeply grateful for the interesting articles suggested by the reviewer about natural agents. This information has been added to the manuscript in the introduction section.

  • It will be more interesting to include a detailed table of target agents and the mechanism of Hepatocellular carcinoma immune surveillance:

Table 1 has been now modified, adding a column describing the mechanism of immune surveillance, as better explained in the test.

  • It will be better to discuss about drug resistance in Hepatocellular carcinoma immune surveillance:

A paragraph about ICI resistance has been added in the introduction section, and this aspect has also been mentioned in the paragraph about antigen recognition.

  • The manuscript needs critical revision and correct grammatical mistakes:

The manuscript underwent critical revision, and grammar mistakes have been identified and corrected.

  • The author targets SerpinB3very often during the study. Could the author through light on another target Hepatocellular carcinoma immune surveillance?:

In the paragraph entitled “Immune System in HCC” different target antigens have been considered in the sub-paragraph entitled “Antigen recognition”, while the various strategies to boost the antitumoral activity of the tumour microenvironment have been described in the following sub-paragraph, as summarized in Table 1. Due to the novelty of the data involving the serin protease inhibitor SerpinB3 as immunomodulator in HCC, we have provided a dedicated paragraph to describe the recent findings addressed to the role of this serpin in the impairment of immune surveillance, adding also a new reference [119] providing evidence that this effect could be mediated via STAT-dependent secretion of chemokine recruiting MDSC and M2 macrophages.

Reviewer 2 Report

This is a timely and interesting review on a rapidly evolving topic.

The authors provide a good summary of key aspects related to the tumor microenvironment (TME) in hepatocellular carcinoma (HCC), focusing on key immune cells and pathways. The relevance of these aspects towards ICI-based therapy of advanced HCC is properly discussed. Finally, the authors discuss the role of SerpinB3 in the regulatoion of TME and immune responses in cancer. The manuscript is well-written and provides a useful summary of the current situation in the field. I only have some suggestions for the authors that can improver the quality of this otherwise commendable manuscript.

1.    It would be helpful to know/discuss if SerpinB3 expression is related to the immune/inflammatory landscape of HCCs.

2.    Are the discussed biological activities of SerpinB3 unique to this serpin family mamber? Are these activities shared by other structurally related serpins?

3.    What are the known mechanisms underlying SerpinB3 activity on immune cells (and tumor cells)? Are there cell surface receptors for SerpinB3?

4.    How do authors envision the pharmacological targeting of SerpinB3 in HCC?

5.    Please change “Druvalumab” for Durmalumab.

6.    Reference 126 seems to be incomplete. Please check the references list. 

Author Response

We appreciate the overall consideration of the reviewer for our manuscript and his suggestions.

The specific points are addressed below:

  • It would be helpful to know/discuss if SerpinB3 expression is related to the immune/inflammatory landscape of HCCs:

In the paragraph “Role of SerpinB3 in the immunosurveillance of HCC” we have now expanded the findings about the role of SerpinB3 in the immune landscape. The activation of TGF-beta and beta-catenin pathways by this serpin can have a direct effect in the composition of the immune infiltrate, as previously described in the NASH models [125, 126]. However, also the recruitment of MDSC and M2 macrophages could be an important mechanism mediated by SerpinB3 [119].

  • Are the discussed biological activities of SerpinB3 unique to this serpin family member? Are these activities shared by other structurally related serpins?

Unfortunately, we don’t know for sure if the discussed biological activities of SerpinB3 are shared with other serpin family members and this hypothesis is intriguing and worth to explore. However, we added in the sub-paragraph “TGF-beta and WNT-beta-catenin pathway activation” the information that the activation of Wnt pathway is mediated by a particular cellular membrane binding site which is shared by other SERPIN family members, like alpha1-antitrypsin and nexin-1 [125].

  • What are the known mechanisms underlying SerpinB3 activity on immune cells (and tumor cells)? Are there cell surface receptors for SerpinB3?

SerpinB3 has been proven to have a paracrine activity and can stimulate cytokines secretion, that is dependent on its anti-protease activity [126], likely acting through a surface receptor, whose characterization is currently under investigation in our laboratory. In addition, recent findings have shown that SerpinB3 determines the activation of the Wnt canonical pathway and cell invasiveness through the upregulation of LRP co-receptors family members [125] and this information has now been added in the sub-paragraph “TGF-beta and WNT-beta-catenin pathway activation”.

  • How do authors envision the pharmacological targeting of SerpinB3 in HCC?

As reported in the conclusion statement, we believe that pharmacological targeting of SerpinB3 could be a novel approach in HCC.  Targeting both the reactive site loop of this serpin and/or blocking its interaction with the LRP co-receptors could be new strategies for HCC treatment.

Minor points:

  • Please change “Druvalumab” for Durvalumab: we have corrected the mistake.
  • Reference 126 seems to be incomplete. Please check the references list: reference 126 has been completed.

Round 2

Reviewer 1 Report

the author revises the manuscript accordingly. the needs to revise and check the final version of the manuscript. overall I accept the manuscript.